# Extracellular Vesicle Abundance, but Not a High Aggregation-Prone Peptide Cargo, Is Associated with Dihydroartemisinin Exposure in *Plasmodium falciparum*

**DOI:** 10.3390/ijms26093962

**Published:** 2025-04-22

**Authors:** Kwesi Z. Tandoh, Yunuen Avalos-Padilla, Prince Ameyaw, Elisabeth K. Laryea-Akrong, Gordon A. Awandare, Michael David Wilson, Neils B. Quashie, Xavier Fernàndez-Busquets, Nancy O. Duah-Quashie

**Affiliations:** 1West African Centre for Cell Biology of Infectious Pathogens, Department of Biochemistry, Cell and Molecular Biology, College of Basic and Applied Sciences, University of Ghana, Legon, Accra P.O. Box LG 54, Ghana; eklaryea-akrong001@st.ug.edu.gh (E.K.L.-A.); gawandare@ug.edu.gh (G.A.A.); nbquashie@ug.edu.gh (N.B.Q.); nduah@noguchi.ug.edu.gh (N.O.D.-Q.); 2Department of Epidemiology, Noguchi Memorial Institute for Medical Research, College of Health Sciences, University of Ghana, Legon, Accra P.O. Box LG 54, Ghana; prinameyaw@noguchi.ug.edu.gh; 3Institute for Bioengineering of Catalonia (IBEC), The Barcelona Institute of Science and Technology (BIST), Baldiri Reixac 10-12, 08028 Barcelona, Spain; yavalos@ibecbarcelona.eu; 4Barcelona Institute for Global Health (ISGlobal, Hospital Clınic-Universitat de Barcelona), Rosselló 149-153, 08036 Barcelona, Spain; 5Department of Parasitology, Noguchi Memorial Institute for Medical Research, College of Health Sciences, University of Ghana, Legon, Accra P.O. Box LG 54, Ghana; 6Centre for Tropical Clinical Pharmacology and Therapeutics, School of Medicine and Dentistry, College of Health Sciences, University of Ghana, Legon, Accra P.O. Box LG 54, Ghana

**Keywords:** artemisinin resistance, extracellular vesicles abundance, dihydroartemisinin exposure, *Plasmodium falciparum*, aggregated proteins, EV exports hypothesis

## Abstract

Our understanding of the molecular mechanisms undergirding artemisinin (ART) resistance in *Plasmodium falciparum* is currently based on two organizing principles: reduced hemoglobin trafficking into the digestive food vacuole, resulting in lower levels of activated ART, and increased tolerance to ART-induced oxidative stress in the infected erythrocyte. We had previously proposed an extracellular vesicle (EV) export model of ART resistance in *P. falciparum*. This model predicts that EV abundance will be altered by ART exposure and that the peptide cargo of EVs from the ART-exposed condition will be enriched with aggregation-prone peptides. We tested the predictions of the EV export hypothesis in this study using in vitro culture assays of an ART-resistant transgenic line engineered on a 3D7 background (R561H) and a 3D7 knock-out line (PfVps60KO) with deficient EV production phenotype. EV enrichment was obtained from in vitro parasite culture supernatants via a series of ultracentrifugation and filtration steps, followed by size exclusion chromatography. A quality check on EVs was performed using dynamic light scattering. Liquid chromatography with tandem mass spectrometry was used to determine the proteome cargo from extracted EVs, and parasite peptides were queried for aggregation-prone tendency using open-access software. We report that dihydroartemisinin (DHA) exposure was positively correlated with EV abundance (coefficient estimate = 1038.58, confidence interval of 194.86–1882.30, and *p*-value = 0.018) and suggests that EV biogenesis is part of the parasite’s response to DHA/ART. Furthermore, our findings suggest the expression of a non-constitutive DHA-induced alternate EV biogenesis pathway as the PfVps60KO was observed to produce the highest number of EVs under DHA exposure. Finally, we show that EVs from both ART-susceptible and resistant parasites under DHA exposure carry a cargo of Chorein N-terminal domain-containing protein (PF3D7_1021700) with a high aggregation-prone index (prion-like domain [PrLD] score = 26.5) out of nine identified parasite peptides. The former of these findings is in concordance with the EV export hypothesis, which posits that the removal of DHA/ART-induced aggregated and/or misfolded peptides is critical to the parasite’s survival under DHA/ART exposure. This observation further implicates EVs in the development of the ART-resistant phenotype. However, the finding of one aggregation-prone peptide out of the nine parasite proteins in the EV cargo does not sufficiently support the EV export hypothesis. Future replicates of this study and further interrogations of the EV export hypothesis are needed.

## 1. Introduction

Since the evolution of artemisinin (ART) resistance in *Plasmodium falciparum* parasites of Southeast Asia (SEA), the risk of spread to sub-Saharan Africa (sSA) has become a significant concern for global malaria eradication efforts [1,2,3]. Additionally, the de novo emergence and clonal expansion of ART-resistant *P. falciparum* parasites in Rwanda and Uganda add further impetus to this problem [4,5]. To mitigate this risk, deepening our understanding of the molecular mechanisms underpinning ART resistance in *P. falciparum* parasites is needed. Such knowledge will drive translational agendas for novel antimalarial drug discovery efforts and improve existing molecular surveillance approaches [6,7,8].

Our knowledge and understanding of ART resistance has primarily centered on the *P. falciparum* kelch 13 (*PfK13*) gene. This *PfK13* model of ART resistance highlights two components: reduced endocytosis of hemoglobin into the digestive food vacuole and increased tolerance to ART-induced oxidative stress [6]. Extracellular vesicles (EVs) are a group of heterogeneous membrane-delimited biological particles implicated in multiple roles in the biology of *P. falciparum* [9]. One of the pathways described for EV biogenesis in *P. falciparum* is achieved by the endosomal sorting complex required for transport (ESCRT)-III complex, which is activated by an alternative pathway for the recruitment of PfBro1 and PfVps32/Vps60 [10]. A role for EVs in the molecular mechanisms undergirding *P. falciparum* ART resistance has traditionally been explored within the hypothetical parasite–parasite communication framework [11]. We have suggested a novel framework called the EVs export hypothesis, outside this zeitgeist of parasite-parasite communication, for interrogating the role of EVs in developing the ART resistance phenotype [12].

The EV export hypothesis posits that ART resistance emerges in parasites that can withstand ART-driven oxidative and proteotoxic assault by actively removing damaged proteins from the infected erythrocyte to maintain cellular homeostasis and proteostasis under ART pressure. We suggest that this proteinopathic export is achieved through EVs generated by the *P. falciparum*-infected erythrocyte. Therefore, the EV export model of ART resistance predicts that EV numerical abundance will be altered in parasites under ART pressure and that the peptide cargo of EVs from ART-exposed parasites would be enriched for misfolded, poly-ubiquitylated, and aggregation-prone peptides.

To test some predictions of this hypothesis, we exposed the *P. falciparum* lines R561H (ART-resistant) and PfVps60KO (reduced EV abundance and ART-susceptible) and their parental 3D7 strain to 0.1% DMSO and 700 nM dihydroartemisinin (DHA). We measured the EV abundance for each parasite line under each condition and determined the peptide cargo of the enriched EVs using liquid chromatography with tandem mass spectrometry. We report that DHA treatment is positively correlated with EV abundance (coefficient estimate = 1038.58, confidence interval of 194.86–1882.30, and *p*-value = 0.018) and that EVs from both ART-susceptible and resistant parasites under DHA exposure carry a cargo of Chorein N-terminal domain-containing protein (PF3D7_1021700) with a high aggregation-prone index (prion-like domain [PrLD] score = 26.5). The former finding suggests that EV biogenesis is part of the parasite’s response to ART and provides evidence to support the EV export hypothesis of ART resistance in *P. falciparum* parasites. The latter finding, juxtaposed to the parasite’s compendium of largely aggregation-prone peptides, is not an unequivocal observation in concordance with the EV exports hypothesis and warrants further exploration.

## 2. Results

### 2.1. Determination of Parasite Sensitivity to Artemisinin

As expected, the R561H line was ART-resistant, with a mean ring-stage survival analysis (RSA) survival rate of 12.7% (standard deviation (sd) of 2.4), whereas the 3D7 and PfVps60KO lines were ART-susceptible, with mean RSA survival rates of 0.7% (sd = 0.08) and 0.4% (sd = 0.09), respectively (Table 1, Appendix A).

### 2.2. Size Distribution of EVs Is Similar Among Parasite Lines for All Conditions

We determined whether the size distributions of the purified EVs were similar between parasite and treatment conditions. This was to ensure validity in the comparison of the derived EV abundance count between parasite lines and treatment conditions. Therefore, we controlled for in vitro culture parasitemia, dilution factors during EV extraction, the duration of EV counting during the dynamic light scattering analysis, and EV size distribution (Figure 1, Appendix A). By controlling for EV size distribution, we effectively accounted for the effect of infected erythrocyte death and cellular debris as confounders of EV abundance. The median EV size was 215.6 nm (first quartile, Q1 = 197.1 nm; third quartile, Q3 = 229.3 nm). EV size distribution did not differ among the parasite lines and treatment conditions. The mean EV sizes were 212 ± 30, 211 ± 42, and 214 ± 28 nm for the 3D7, PfVps60KO, and R561H lines, respectively (Figure 2, Appendix A).

### 2.3. DHA Treatment Condition Is the Best Predictor of EV Abundance

The EV abundance was greatest in the 3D7 parasite line compared to the PfVps60KO and R561H parasite lines for both the untreated and DMSO-treated conditions. However, for these control conditions, a statistically significant difference in EV abundance was only seen in the DMSO-treated condition (Table 2). EV abundance in the DHA-treated condition was highest in the PfVps60KO parasite line but showed no statistically significant differences in comparison to the 3D7 and R561H parasite lines (Table 2 and Appendix A). Under the DMSO-treated condition, only the 3D7 parasite line compared to the PfVps60KO parasite line showed a statistically significant difference in EV abundance (Appendix A).

We expected that comparisons between the 3D7 and PfVps60KO parasite lines, for the untreated and DMSO-treated conditions, would show statistically significant differences with the greater EV abundance seen in 3D7 [10]. However, the lack of statistical significance in the EV abundance between 3D7 and PfVps60KO for the untreated condition may be explained by the presence of outliers in the technical replicates and their contribution to within-sample variance (Appendix A). This expected difference in EV abundance between 3D7 and PfVps60KO was seen with marginal statistical significance in the DMSO-treated control arm (Table 2, Appendix A).

Comparisons of treatment conditions showed that EV abundance was greatest in the DHA-treated condition compared to the untreated and DMSO-treated conditions (Table 2, Appendix A). A statistically significant difference in EV abundance was only seen in the PfVps60KO and R561H parasite lines for the DHA-treated condition compared with the DMSO-treated condition (Table 2, Appendix A, Appendix A). In the PfVps60KO parasite line, EV abundance in the DMSO-treated condition was lower compared to the 3D7 parasite line (Table 2, Appendix A, Appendix A) and was highest in the DHA-treated condition compared to the 3D7 and R561H parasite lines (Table 2, Appendix A, Appendix A).

As the similar genetic background across the used parasite lines implied a shared EV biogenesis machinery, it was expected that there would be no statistically significant differences in EV abundance between the 3D7 and R561H parasite lines for both the untreated and DMSO-treated conditions (Table 2, Appendix A, Appendix A). Additionally, as the DMSO-treated condition was used as a solvent/vehicle control for the DHA-treated condition, a statistically significant difference in EV abundance between the untreated and DMSO-treated condition was not expected (Table 2, Appendix A, Appendix A). We did expect and found differences in EV abundance between the DHA-treated and DMSO-treated conditions for the PfVps60KO and R561H parasite lines (Appendix A, Appendix A).

Finally, we considered the independent variable parasite lines (3D7, PfVps60KO, and R561H) and treatment condition (untreated, DMSO-treated, and DHA-treated) as predictors of the response variable, EV abundance. All these factors were used as predictor variables in a multivariate simple linear regression model to determine the strength and statistical significance of their association with EV abundance. We found that the DHA-treated condition was positively correlated with EV abundance (coefficient estimate = 1038.58, confidence interval of 194.86–1882.30, and *p*-value = 0.018) (Table 3, Figure 3).

### 2.4. One Aggregation-Prone Peptide out of Nine Identified Peptides Was Found in the Proteome Cargo of EVs from DHA-Exposed ART-Susceptible and Resistant Parasites

We interrogated the proteome cargo of EVs from the three parasite lines under the three treatment conditions with the aim of determining whether their cargo contained aggregated peptides. First, from the mass spectrometry analysis of the 27 samples (sixty-four injections, at least two injections per sample), 156 proteins were identified (FDR < 5%, at protein level) by Proteome Discoverer (v. 2.5). A total of 140 proteins remained after excluding proteins that had no peptides identified in at least two sample injections (Appendix A). From this subset, nine were identified as *P. falciparum* proteins (Appendix A). We then determined which of the nine proteins was enriched in the DHA-treated group in comparison to the untreated and DMSO-treated groups. Only the Chorein N-terminal domain-containing protein (PF3D7_1021700) had higher abundances in the DHA-treated group compared to the DMSO-treated and untreated groups. Furthermore, PF3D7_1021700 showed a reduced abundance in the PfVps60KO parasite line under DHA treatment even though EV abundance from this parasite line under DHA treatment was the most abundant (Appendix A). Finally, we determined the aggregation-prone tendency of these nine proteins by measuring the proportion of prion-like domains in their peptide sequences using the Prion-Like Amino Acid Composition (PLAAC) tool [13,14]. PF3D7_1021700, the only parasite peptide abundant under the DHA-treated condition in the 3D7 and R561H parasite lines, had a high aggregation-prone index (prion-like domain (PrLD) score = 26.5) (Table 4).

## 3. Discussion

This study determined the association between parasite lines and treatment conditions, and EV abundance, in *P. falciparum*. The DHA-treated condition was associated with EV abundance in a linear regression model. This finding suggests the induction by DHA of an alternative EV biogenesis pathway in *P. falciparum* and finds concordance with previous reports of heterogeneity in EV biogenesis pathways [15,16]. The association between EV numerical abundance and malaria has been reported for malaria parasitemia and cerebral malaria [17,18,19]. However, the association between EV abundance and proteomic cargo, and the ART-resistant phenotype or DHA-treated condition, has not been previously explored. This captures the novel contribution of our findings to the existing body of knowledge on ART resistance in *P. falciparum*.

Our finding that DHA is positively correlated with EV abundance implicates a role for the EV biogenesis pathway in the *P. falciparum* parasite’s response to DHA in vivo. DHA action is known to involve the alkylation of parasite proteins, the induction of a total proteotoxic state, and the activation of the unfolded protein response [12]. We hypothesized that, under this DHA assault, EV biogenesis is induced to execute the removal of accumulating aggregated, alkylated, and ubiquitinated proteins to ensure proteostasis. Thus, our finding of increased EV abundance in all parasite lines under the DHA-treated condition supports the plausibility of the EV export model of ART resistance in *P. falciparum*. This is the hypothetical conceptual framework that explains the observations we made. It is the expectation of the EV export hypothesis that EVs under the DHA-treated condition would be most abundant in the ART-resistant parasite line R561H and least abundant in the PfVps60KO parasite line in comparison to the 3D7 parasite line. This was not seen. Instead, under the DHA-treated condition, EVs were most abundant in the PfVps60KO parasite line. In the same vein, EV abundance under the DHA condition was comparable for both the ART-sensitive 3D7 and ART-resistant R561H parasite lines. These observations suggest a DHA-inducible non-constitutive arm to EV biogenesis in *P. falciparum*.

Secondly, the EV export hypothesis predicts that the cargo of EVs from malaria parasites under DHA exposure would be enriched with aggregated proteins and that for an ART-resistant parasite line, this cargo of aggregated proteins would be higher under DHA exposure compared to the DHA-untreated condition [12]. Our findings did not provide evidence to unequivocally support these expectations, as there was a deviation from the expectations of the EV export hypothesis with the observation of the independence of EV abundance to ART susceptibility (Table 3, Figure 3). Chorein N-terminal domain-containing protein (PF3D7_1021700) was the only *P. falciparum* peptide enriched in the DHA-treated conditions for both the 3D7 and R561H parasite lines. Significantly, an analysis of its prion-like domain content showed a high index of aggregation-prone propensity as it contained amino acids biased for glutamine/asparagine (Q/N) repeats. However, given that this was the only aggregation-prone peptide out of the nine peptides in the EV proteomic cargo identified, it is not a cogent argument for enrichment as it is known that up to 10% of the proteome of *P. falciparum* is aggregation-prone [20]. Therefore, although this observation is partially accommodated by the EV export hypothesis, which predicts an enrichment for aggregated peptides in the EV cargo from ART-resistant parasites under DHA exposure, it might also be explained by the baseline constitution of aggregation-prone peptides in *P. falciparum*. We observed this for both the ART-resistant R561H line and the ART-sensitive 3D7 parasite line, which suggests an inducible non-constitutive EV biogenesis pathway associated with DHA exposure as a likely mitigating response in keeping with the expectations of the EV export hypothesis. This finding notwithstanding, we can only cautiously state that the cargo of EVs from DHA-exposed parasites contain aggregation-prone peptides. What remains to be elucidated are the dynamics of EV cargo with the duration of exposure to DHA and how EV export under ART exposure affects the viability of *P. falciparum*.

It can be further argued that compared to the backdrop of the known abundance of aggregation-prone peptides in *P. falciparum* and their seemingly innocuous effect on parasite viability, it is possible that even under the influence of DHA, an EV-exporting DHA-mitigating system cannot significantly alter the abundance of the parasite’s misfolded protein cargo [14,20,21]. Although this is a plausible argument, it remains to be empirically demonstrated by future studies. Additionally, we recognize the uncertainty in our assumption of toxicity of this misfolded proteome cargo under DHA pressure as being central to the expectations of the EV export hypothesis. However, we reason that these arguments do not utterly dismiss the EV export hypothesis. Instead, they raise another important question: can EV export of misfolded or aggregation-prone peptides serve as a means of parasite–parasite communication to aid *P. falciparum* in surviving DHA exposure? Future studies that explore these unanswered questions are needed.

We recognize the following limitations of our study. Although our experimental design and current knowledge on EV biogenesis in *P. falciparum* did not allow us to account for EV biogenesis pathways and differentiate exosomes from microvesicles, we are confident in the robustness of our experimental design that showed the enrichment of EVs of similar average size across parasite lines for all conditions. Additionally, the quality check for purified EVs would have been improved by using transmission electron microscopy and/or Western blot for the known peptide markers of EVs. Our assumption of a linear relationship between EV abundance and parasitemia, parasite lines, and treatment conditions was made a priori and used downstream. Additionally, our study design would have benefitted from more technical replicates of the sample conditions and another antimalarial drug control. Our sample size may have hindered the power of our analysis to determine the strength of the association between DHA exposure and EV abundance. This notwithstanding, our findings lay the groundwork for future studies to improve their sample size to achieve adequate statistical power. Our experimental design for interrogating the proteomic cargo of EVs was limited by the small amounts of proteins extracted from EVs (Appendix A), and on account of this, we could not use the highly sensitive Proteostat dye [14] to interrogate the proteome cargo. This limitation will be remedied in future studies by extracting EVs from larger volumes of parasite culture supernatant. Another limitation to our study is the intrinsic sensitivity of mass spectrometry to detect aggregation-prone peptides. Biosca and others worked around this by employing an alternate assay for investigating aggregation-prone peptides by harvesting peptides that resisted dissolution in a sodium dodecyl sulfate (SDS)-based assay [14]. Future replications would employ this alternative approach as well. We also recognize the limitations in the performance of the PLAAC tool for prion-like domain (PrLD) prediction, which requires independent wet-lab validation of its findings using functional assays. Additionally, future studies will test further the predictions of the EV export hypothesis by determining whether EVs from malaria parasites under DHA exposure carry a peptide cargo of ubiquitylated and alkylated peptides associated with oxidative stress.

Finally, we contextualize our findings that lacked statistical significance by highlighting the argument that the frequentist *p*-value used in null hypothesis significance testing (reject or do not reject) is disadvantaged by the arbitrary threshold of 0.05 or 0.1 used. A *p*-value below 0.05 is considered significant and a *p*-value above 0.05 is not significant even if the difference between them is minimal. This oversimplifies significance decision making as a small *p*-value does not conclusively mean that the effect is practically significant and only indicates that the observed data are unlikely under the null hypothesis. The converse argument holds too, as a large *p*-value does not conclusively mean that the effect is not significant but only indicates that the observed data are more likely under the null hypothesis. Future studies will control for these concerns by using larger sample sizes and performing statistical hypotheses testing using both the frequentist *p*-value and Bayesian statistics Bayes factors.

## 4. Methods

### 4.1. P. falciparum In Vitro Parasite Culture and Treatment Conditions

Asexual stages of the *P. falciparum* lines R561H (ART-resistant) and PfVps60_KO (EV biogenesis-deficient line) and their parental 3D7 strain were cultured in vitro in group B human erythrocytes at 3% hematocrit using complete Roswell Park Memorial Institute (RPMI) medium (Life Technologies, Paisley, UK) supplemented with 0.5% (*w*/*v*) Albumax II (Life Technology, Auckland, New Zealand) and 200 mM L-glutamine (Thermo Fisher Scientific, Waltham, MA, USA). Parasitemia was maintained between 1 and 5% at 37 °C under an atmosphere of 5% O_2_, 5% CO_2_, and 90% N_2_. Prior to EV extraction, parasite cultures were synchronized in the ring stage with a 5% sorbitol lysis (Merck KGaA, Darmstadt, Germany) [22]. Subsequently, 36 h post-invasion (hpi), parasites were synchronized in the schizont stages by 70% Percoll (GE Healthcare, Uppsala, Sweden) treatment. Finally, a second 5% sorbitol lysis was performed after 48 or 96 h to yield ring-stage parasites synchronized at 0–8 hpi (Figure 1). Parasite cultures were then exposed to 700 nM dihydroartemisinin (DHA) (Tebubio, Le Perray-en-Yvelines, France) or 0.1% dimethyl sulfoxide (DMSO) (Merck, KGaA, Darmstadt, Germany) for 6 h (42–48 hpi) prior to harvesting the culture supernatant at 48 hpi for EV extraction (Figure 1). The choice of 700 nM DHA concentration was based on its use in the well-established in vitro ring-stage survival assay. The final culture parasitemia was at least 1% for all conditions except for the DHA-treated samples (Appendix A).

### 4.2. Ring Stage Survival Analysis

The 0–3 hpi ring-stage survival analysis (RSA_0–3_) was performed as described previously [23,24]. In brief, synchronized 0–3 hpi rings at 1% parasitemia and 2% hematocrit were exposed to 700 nM DHA (Tebubio) for 6 h, followed by the removal of the drug. At 72 h after initial drug treatment, parasitemia was assessed with microscopy analysis of Giemsa-stained thin smears. The ring-stage survival percentage was calculated as the fraction of surviving DHA-treated parasites relative to the DMSO-treated control of the same parasite line. The ring-stage survival percentage read out was only interpreted when the parasite growth rate, defined as the non-exposed parasitemia at 72 h divided by initial parasitemia, was greater than 1.5%. An RSA survival percentage > 1% was considered to indicate ART resistance. The RSA_0–3_ for each parasite line was determined, with at least three technical replicates per biological sample.

### 4.3. EV Extraction and Purification

For the purification of EVs, parasite culture supernatants were collected from in vitro *P. falciparum* cultures. EV extraction and purification were carried out as previously described [25]. Briefly, the samples were initially prepared by sequential centrifugations to remove large aggregates. First, the cultures were centrifuged for 10 min at 400× *g*, the cell pellet was discarded, and the resulting supernatant was further centrifuged twice at 2000× *g* for 10 min. In both cases, a small pellet was discarded. Finally, 25 mL of the supernatant was placed in an Amicon Ultra-15 centrifugal filter (100 kDa cut-off, Millipore-Merck, Cork, Ireland) and centrifuged for 10 min at 3000× *g*. One mL of the resulting concentrated solution was collected and transferred to a 10 mL homemade Sepharose CL-4B column (Cytiva, Uppsala, Sweden) previously equilibrated with phosphate-buffered saline (PBS). EV purification was performed with gravity flow at room temperature and 0.5 mL fractions were collected. EVs were enriched in fractions 7, 8, 9, and 10, which were combined, concentrated in an Amicon Ultra-4 centrifugal filter (100 kDa cut-off, Millipore-Merck), centrifuged for 10 min at 5000× *g*, and re-suspended in a final volume of 200 μL to be used for EV abundance determination assays [10] (Appendix A, Figure 1).

### 4.4. EV Size and Abundance Measurements

Dynamic light scattering was used to measure particle size in the purified EV population as previously described [26]. To obtain the optimum light scattering intensity, 100 μL of purified EVs were resuspended in 900 μL of filtered (0.22 μm) PBS diluted 1:2 in 4% paraformaldehyde. The mean particle size of vesicle dispersions and the derived count rate (kilo counts per second) were determined in triplicates from light diffusion measured at 25 °C and an attenuator index of 8, using a Zetasizer Nano S (Malvern Instruments, Ltd., Malvern, UK) [10]. The mean EV abundance was derived from the count rate and duration of the count and normalized for parasitemia and dilution factors used in the experimental design (Appendix A, Figure 1).

### 4.5. Sample Preparation for Mass Spectrometry

Briefly, approximately 10 µg of each sample was added to 8M urea in 50 mM ammonium bicarbonate buffer. Next, the samples were reduced with 18 mM dithiothreitol for 60 min at 32 °C and alkylated with 34 mM iodoacetamide at 25 °C for 30 min in the dark. Subsequently, the samples were digested overnight (pH 8, 32 °C) with trypsin (sequencing grade modified, Promega Corporation, Madison, WI, USA). Finally, the resulting peptide mixtures were acidulated with formic acid, concentrated in a SpeedVac vacuum system, and cleaned up with a C18 spin tip (Thermo Fisher Scientific, IL, USA) in accordance with the manufacturer’s protocol. Finally, the cleaned-up peptide solutions were dried and stored at −20 °C.

### 4.6. Analytical Procedure and Equipment Used for LC–MS/MS

The dried-down peptide mixture was analyzed in a nanoAcquity liquid chromatographer (Waters Corporation, Milford, MA, USA) coupled to an LTQ-Orbitrap Velos (Thermo Fisher Scientific) mass spectrometer. The tryptic digests were resuspended in 1% FA solution and an aliquot per sample was injected for chromatographic separation. The peptides were trapped on a Symmetry C18TM trap column (5 µm, 180 µm × 20 mm; Waters Corporation) and were separated using a C18 reverse phase capillary column (ACQUITY UPLC BEH column; 130 Å, 1.7 μm, 75 μm × 250 mm, Waters Corporation). The A:B gradient used for the elution of the peptides was 1 to 40% B in 30 min, followed by a gradient from 40% to 60% B in 5 min. In addition, samples 1 to 9 were also analyzed using a gradient from 1 to 40% B in 60 min, followed by a gradient from 40% to 60% B in10 min (A: 0.1% FA I water; B: 0.1% FA in acetonitrile; flow rate: 250 nL/min). The column temperature was 40 °C. Eluted peptides were subjected to electrospray ionization in an emitter needle (Thermo Fisher Scientific) with an applied voltage of 2100 V. The peptide masses (*m*/*z* 30–1600) were analyzed in data-dependent mode, where a full-scan MS was acquired in the Orbitrap with a resolution of 60,000 FWHM at 400 *m*/*z.* Up to the 15th most abundant peptides (minimum intensity of 500 counts) were selected from each MS scan and then fragmented in the linear ion trap using CID (38% normalized collision energy) with helium as the collision gas. The scan time settings were full MS: 250 ms (1 microscan) and MSn: 120 ms. The generated raw data files were collected with ThermoXcalibur (v.2.2).

### 4.7. Data Analysis for LC–MS/MS Data

Briefly, database searching and protein identification were performed by using the raw data files obtained in the mass spectrometry analyses as queries to search against a modified version of the public database, Uniprot, for *P. falciparum* 3D7 (v.11/12/23) with all entries for *Homo sapiens* present in the database SwissProt (v. 11/12/23). A small database containing laboratory contaminant proteins was also added. The database search was performed with SequestHT search engine using Thermo Proteome Discover (v.2.5). The following search parameters were applied: carbamidomethylation of cysteines as fixed modifications, methionine (M) oxidation and acetylation, Met-loss, and Met-loss-acetylation (N terminus) as variable modifications. The enzyme used was trypsin with a maximum allowed missed cleavage of 2. The peptide tolerance was set to 20 ppm and 0.6 Da (respectively, for MS and MS/MS spectra). The false discovery rate (FDR) was set to 1% for peptide spectrum match (PSM) levels.

### 4.8. Statistical Analysis

The tests for normality of numerical data distribution were performed using the Shapiro test. The Bartlett test was used to determine the equality of variances between comparator groups. The Kruskal–Wallis test was used to compare EV numerical abundance between comparator groups. Adjusted *p*-values were determined using the Dunn post hoc test and adjusted with the Benjamini–Hochberg method. Data analysis was performed using custom R scripts in R version 4.0 [27].

## 5. Conclusions

In conclusion, we report that DHA exposure is positively correlated with EV abundance and rescues the wild-type EV biogenesis phenotype in PfVps60KO. This suggests that EVs play a role in the parasite’s response to DHA/ART and in the development of the ART resistance phenotype. Secondly, we found that only one aggregation-prone peptide, albeit in the DHA-treated condition, out of the nine parasite peptide cargo does not make for a cogent unequivocal observation in concordance with the EV export hypothesis. It does, however, underscore the implication of EVs as probable vehicles of proteome export as suggested by the EV export hypothesis.

## Figures and Tables

**Figure 1 ijms-26-03962-f001:**
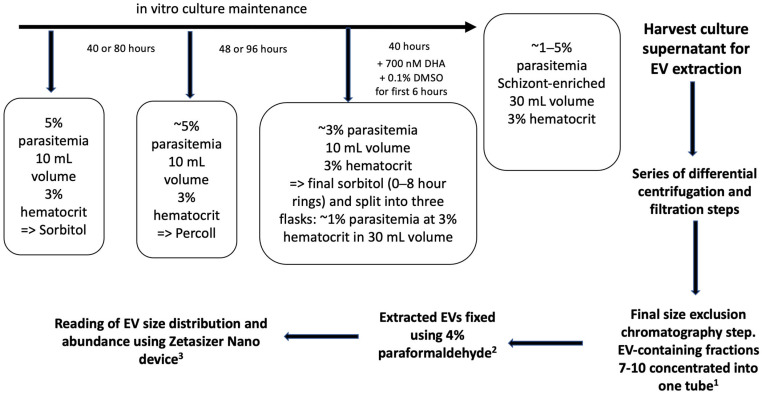
Workflow for extracellular vesicle (EV) extraction, purification, and abundance estimation. Parasite cultures were synchronized to 8 h rings and the EVs were harvested from the culture supernatant after 40 h. A series of centrifugation and filtration steps were used to extract the EVs. Finally, a sepharose chromatography column was used to enrich EVs in fractions 7 to 10. Fractions 7–10 were then concentrated using a final filtration step and the final volume was adjusted to 200 µL with 1× PBS ^1^. After this, EVs were fixed using 4% paraformaldehyde in a 1:2 dilution ^2^. EV size and abundance were then estimated with a Zetasizer Nano device. A 1:9 dilution with 1× PBS was used for loading the fixed EVs onto the Zetasizer Nano device. ^1^: first dilution factor; ^2^: second dilution factor; ^3^: third dilution factor.

**Figure 2 ijms-26-03962-f002:**
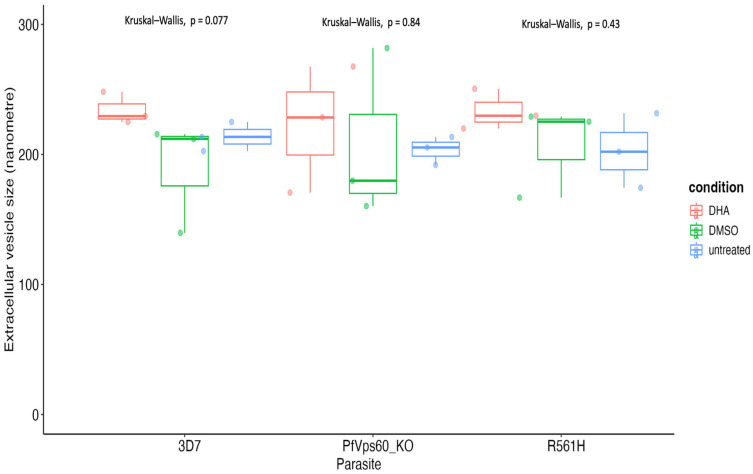
Analysis of EV size for the different parasite lines and treatments. The comparison of EV size among parasite lines and treatments was performed using the Kruskal–Wallis test. The comparison of EV size among the three parasite lines showed no differences in size distribution (Kruskal–Wallis *p*-value = 0.74).

**Figure 3 ijms-26-03962-f003:**
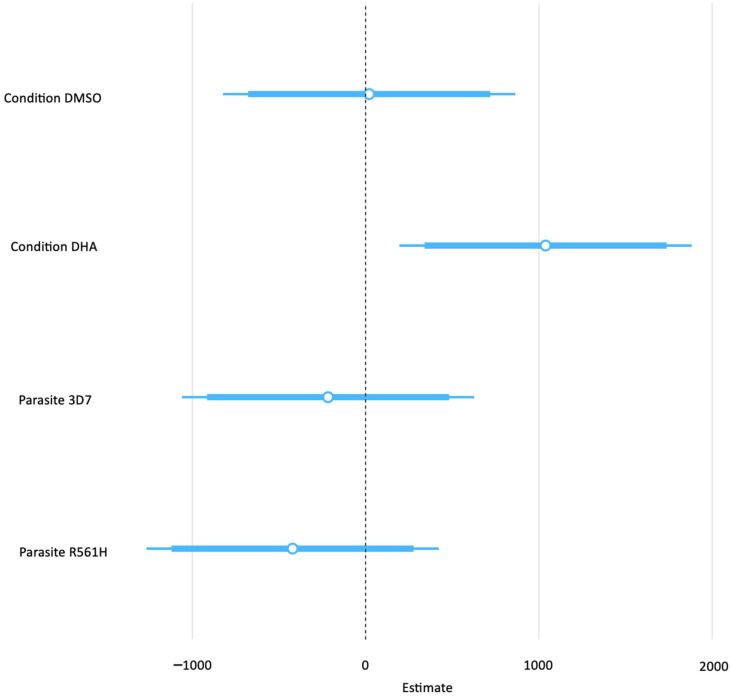
Strength of the association between the predictor factors’ treatment condition and parasite line, as well as the response variable EV abundance treatment condition factors, were compared to the untreated condition baseline; parasite lines were compared to the PfVps60 baseline.

**Table 1 ijms-26-03962-t001:** In vitro 0–3 h post infection ring-stage survival analysis (RSA_0–3_) parasite survival rates (%) to artemisinin (ART) treatment for the *P. falciparum* strains 3D7, R561H, and PfVps60KO. Results represent the mean of triplicate experiments. (**A**) RSA_0–3_ was above 1% for R561H and less than 1% for 3D7 and PfVps60KO. (**B**) Post hoc Dunn’s test for the RSA analysis for 3D7, PfVps60KO, and R561H. *p*-values were adjusted with the Benjamini–Hochberg method.

(A)			
Parasite Line	RSA_0–3_ % (Standard Deviation)	*p*-Value (Kruskal–Wallis Test)	Interpretation
3D7	0.7% (0.08)	4 × 10^−5^	ART sensitive
R561H	12.7% (2.4)		ART resistant
PfVps60KO	0.4% (0.09)		ART sensitive
(**B**)			
**Comparison with Dunn’s Post Hoc Test**	**Z Value**	***p*-Value Unadjusted**	***p*-Value Adjusted**
3D7 versus R561H	−2.302104	2.132930 × 10^−2^	3.199395 × 10^−2^
3D7 versus PfVps60KO	2.195853	2.810245 × 10^−2^	2.810245 × 10^−2^
R561H versus PfVps60KO	4.497957	6.860944 × 10^−2^	2.058283 × 10^−2^

**Table 2 ijms-26-03962-t002:** Analysis of EV abundance by parasite line and treatment condition factors. Field values are medians (first quartile, third quartile). The comparison of EV abundance among parasite lines and treatment conditions was performed using the Kruskal–Wallis test with the degree of freedom = 2.

	EV Abundance (Mega Counts)				
	Untreated (n = 9)	DMSO (n = 9)	DHA (n = 9)	KW Chi-Square	*p*-Value
PfVps60KO (n = 9)	321 (241, 338)	93 (91, 114)	1572 (1222, 3254)	7.2	0.03
3D7 (n = 9)	414 (358, 563)	400 (369, 1030)	700 (692, 1037)	1.16	0.56
R561H (n = 9)	376 (314, 462)	245 (191, 389)	652 (628, 1024)	5.96	0.05
KW chi-square	1.87	5.96	3.47		
*p*-value	0.39	0.05	0.18		

EVs—extracellular vesicles; DMSO—dimethyl sulfoxide; DHA—dihydroartemisinin; KW—Kruskal–Wallis.

**Table 3 ijms-26-03962-t003:** Parameters of a simple linear regression model to determine the strength of the association between parasite lines and treatment conditions (predictor/independent factors), as well as EV abundance (response/dependent factor). The baseline reference to which parasite lines were compared was PfVps60KO, untreated for the treatment conditions.

	EV Abundance (Mega Counts)		
Predictors	Estimates	Confidence Interval	*p*-Value
(Intercept)	595.23	−174.98–1365.44	0.123
Condition DMSO	20.45	−823.27–864.18	0.960
Condition DHA	1038.58	194.86–1882.30	0.018
Parasite 3D7	−216.71	−1060.43–627.01	0.600
Parasite R561H	−421.42	−1265.14–422.30	0.312
Observations	27		
R^2^/R^2^ adjusted	0.304/0.177		

**Table 4 ijms-26-03962-t004:** Aggregation-prone measurement of the nine identified *P. falciparum* proteins using the PLAAC software (http://plaac.wi.mit.edu, accessed on 18 July 2024).

PlasmoDB ID	Protein Name	PrLD Score (PLAAC)
PF3D7_1035300	Glutamate-rich protein GLURP	Undefined
PF3D7_1401400	Early transcribed membrane protein	Undefined
PF3D7_1353200	Membrane-associated histidine-rich protein 2	Undefined
PF3D7_1444800	Fructose-bisphosphate aldolase	Undefined
PF3D7_1149000	Antigen 332, DBL-like protein	Undefined
PF3D7_0727400	Proteasome subunit alpha type	Undefined
PF3D7_1355300	Putative histone-lysine N-methyltransferase 1	Undefined
PF3D7_1021700	Chorein N-terminal domain-containing protein	26.5
PF3D7_1234000	Thioredoxin domain-containing protein	Undefined

## Data Availability

The data presented in this study are available in the Appendix A section.

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
