# Peer review of "Extracellular Vesicle Abundance, but Not a High Aggregation-Prone Peptide Cargo, Is Associated with Dihydroartemisinin Exposure in Plasmodium falciparum"

_ijms, 2025, doi:10.3390/ijms26093962_

Round 1

Reviewer 1 Report

Comments and Suggestions for Authors

The authors describe the increased production of extracellular vesicles (EVs) by different strains of Plasmodium falciparum when treated with DHA, an active derivative of artemisinin (ART). According to the authors' hypothesis, the production of EVs by ART-treated parasites would be related to resistance to ART treatment, and EVs would be a way of exporting misfolded, ubiquitinated or aggregated proteins, increasing parasite survival. Using different strains of P. falciparum, with or without resistance to ART, and treated with DHA, the authors analyzed the production of EVs, verifying their abundance, relative content of peptides, especially those with a tendency to aggregate, relating the results obtained to the EV export hypothesis.

The authors conclude that exposure to DHA is positively correlated with the abundance of EVs even in PfVps60KO parasites, i.e. those that in theory could not produce EVs due to the lack of a gene associated with this production, showing that there are more pathways involved in the production of EVs. This suggests that EVs play a role in the parasite's response to ART and, therefore, in the development of the ART resistance phenotype. In addition, they note that only one aggregation-prone peptide was identified in the parasites' EVs, which does not directly corroborate the EV exhortation hypothesis, but suggests that EVs may be likely vehicles for the export of defective proteins for eventual parasite survival and resistance to ART, as suggested by the EV export hypothesis.

Considering the literature cited, the presented methodology, and the results obtained and discussed, I believe that this is a study with relevant progress in the area of research into the resistance of P. falciparum strains to ART, which is fundamental in the search for new treatments for malaria.

Some considerations regarding the manuscript:

1) Revise Figures 2 and 3, they appear to be low resolution and stretched, leave a better resolution and size for the final version;

2) On line 273, format the references cited in the same way as the rest of the article, in number format, not author/date format.

Author Response

The authors describe the increased production of extracellular vesicles (EVs) by different strains of Plasmodium falciparum when treated with DHA, an active derivative of artemisinin (ART). According to the authors' hypothesis, the production of EVs by ART-treated parasites would be related to resistance to ART treatment, and EVs would be a way of exporting misfolded, ubiquitinated or aggregated proteins, increasing parasite survival. Using different strains of P. falciparum, with or without resistance to ART, and treated with DHA, the authors analyzed the production of EVs, verifying their abundance, relative content of peptides, especially those with a tendency to aggregate, relating the results obtained to the EV export hypothesis.

The authors conclude that exposure to DHA is positively correlated with the abundance of EVs even in PfVps60KO parasites, i.e. those that in theory could not produce EVs due to the lack of a gene associated with this production, showing that there are more pathways involved in the production of EVs. This suggests that EVs play a role in the parasite's response to ART and, therefore, in the development of the ART resistance phenotype. In addition, they note that only one aggregation-prone peptide was identified in the parasites' EVs, which does not directly corroborate the EV exhortation hypothesis, but suggests that EVs may be likely vehicles for the export of defective proteins for eventual parasite survival and resistance to ART, as suggested by the EV export hypothesis.

Considering the literature cited, the presented methodology, and the results obtained and discussed, I believe that this is a study with relevant progress in the area of research into the resistance of P. falciparum strains to ART, which is fundamental in the search for new treatments for malaria.

Some considerations regarding the manuscript:

1) Revise Figures 2 and 3, they appear to be low resolution and stretched, leave a better resolution and size for the final version;

Response: This has been revised accordingly.

2) On line 273, format the references cited in the same way as the rest of the article, in number format, not author/date format.

Response: This has been revised accordingly. Now line 276

Reviewer 2 Report

Comments and Suggestions for Authors

The current article entitled as “Extracellular vesicles abundance, but not a high aggregation-prone peptide cargo, is associated with dihydroartemisinin exposure in Plasmodium falciparum” provides valuable insights into the relationship between extracellular vesicles (EVs) biogenesis and dihydroartemisinin (DHA) exposure in Plasmodium falciparum. The findings reveal a positive correlation between DHA treatment and EVs abundance, suggesting a potential role of EVs in the parasite’s stress response. Although only one aggregation-prone protein was identified in the EVs cargo, the study contributes to the growing understanding of parasite adaptation mechanisms under drug pressure. The research offers a foundation for further exploration of EVs as potential biomarkers or therapeutic targets in malaria treatment. However, before accepting this work, several issues must be addressed, with particular emphasis on the points listed below:

  1. Abstract
    1. The statement "our findings suggest a plurality in the molecular pathways underlying ART resistance" is vague. The term "plurality" is not specific enough for scientific communication.
    2. The phrase "one out of the nine parasite proteins in the EVs cargo found to be aggregation prone does not suggest an enrichment in aggregation prone peptide cargo" is cumbersome and unclear. If only one protein was found, does this mean the hypothesis was largely unsupported?
  2. Introduction
    1. The connection between PfVps60KO and ART susceptibility is underexplored. Why does the loss of PfVps60 reduce EVs production? Is this the only pathway responsible for EVs biogenesis?
    2. The exposure of R561H and PfVps60KO strains to 700 nM DHA is not sufficiently justified.
      1. Why were 700 nM chosen? Is this physiologically relevant?
      2. How does this compare with DHA plasma concentrations in patients receiving artemisinin-based combination therapy?
    3. "A role for EVs in the molecular mechanisms undergirding P. falciparum ART resistance has traditionally been explored within the hypothetical parasite-parasite communication motif."
      1. "Motif" is an inappropriate term in this context. A more precise term would be "framework" or "hypothesis”.
    4. Result
      1. The resistance phenotype determination relies solely on Ring-stage Survival Assay (RSA), but there is no mention of biological replicates beyond triplicates. Given the variability in RSA, triplicates might not be sufficient to draw strong conclusions.
      2. The interpretation of EVs abundance under different conditions is inconsistent:
        1. The claim "DHA treatment condition is the best predictor of EVs abundance" is based on a simple linear regression model with an R² of only 0.304 (adjusted R² = 0.177). This means only ~17% of the variation is explained by the model, which is very weak.
        2. The p-value for DHA treatment (0.018) is modest, and there is no correction for multiple hypothesis testing, which raises the risk of false positives.
  • Why are DHA-treated PfVps60KO parasites producing the most EVs despite being described as deficient in EVs production? This contradicts the premise that PfVps60KO should have impaired EVs biogenesis.
  1. The PLAAC tool for prion-like domain (PrLD) prediction is prone to false positives, and there is no independent validation of aggregation-prone behavior using biochemical assays (e.g., Thioflavin T fluorescence, Congo Red binding, or TEM imaging)
  2. Was the depth of LC-MS/MS proteomics sufficient? Why were only 9 falciparum proteins detected?
  3. The claim that DMSO controls should not show differences in EVs production contradicts observed differences between 3D7 and PfVps60KO in the DMSO condition.
  1. Discussion
    1. The hypothesis predicts that ART-resistant parasites should produce more EVs under DHA to remove damaged proteins. However, the data shows that the ART-resistant strain (R561H) did not have the highest EVs levels. Instead, PfVps60KO, an ART-sensitive strain, had the most EVs under DHA exposure, which does not fit the hypothesis.
  2. Materials and Methods
    1. No functional tests (e.g., assays showing actual protein misfolding or toxicity) were done to confirm that these proteins are harmful or need to be removed via EVs. Electron microscopy (EM) or nanoparticle tracking analysis (NTA) is not used to confirm whether the isolated particles are true EVs or cellular debris. Additionally, there is no mention of using EVs markers (e.g., CD63, TSG101, HSP70) to confirm vesicle identity.
    2. The study relies on PLAAC software to predict aggregation-prone peptides, but no experimental validation was done.
Comments on the Quality of English Language

The manuscript must be checked for English text and grammar.

Author Response

Comments and Suggestions for Authors
The current article entitled as “Extracellular vesicles abundance, but not a high aggregation-prone peptide cargo, is associated with dihydroartemisinin exposure in Plasmodium falciparum” provides valuable insights into the relationship between extracellular vesicles (EVs) biogenesis and dihydroartemisinin (DHA) exposure in Plasmodium falciparum. The findings reveal a positive correlation between DHA treatment and EVs abundance, suggesting a potential role of EVs in the parasite’s stress response. Although only one aggregation-prone protein was identified in the EVs cargo, the study contributes to the growing understanding of parasite adaptation mechanisms under drug pressure. The research offers a foundation for further exploration of EVs as potential biomarkers or therapeutic targets in malaria treatment. However, before accepting this work, several issues must be addressed, with particular emphasis on the points listed below:

Abstract

The statement "our findings suggest a plurality in the molecular pathways underlying ART resistance" is vague. The term "plurality" is not specific enough for scientific communication.

Response: The sentence has been revised accordingly.

Lines 38 - 40 “Furthermore, our findings suggest the expression of a non-constitutive DHA-induced alternate EVs biogenesis pathway, as the PfVps60KO was observed to produce the highest number of EVs under DHA exposure.”

Lines 244 – 246 “These observations suggest a DHA-inducible non-constitutive arm to EVs biogenesis in P. falciparum.”  

The phrase "one out of the nine parasite proteins in the EVs cargo found to be aggregation prone does not suggest an enrichment in aggregation prone peptide cargo" is cumbersome and unclear. If only one protein was found, does this mean the hypothesis was largely unsupported?
Introduction

Response: The sentence has been clarified accordingly to read as " However, the finding of one aggregation prone peptide out of the nine parasite proteins in the EVs cargo does not sufficiently support the EVs export hypothesis". Lines 47 and 48.

Introduction

The connection between PfVps60KO and ART susceptibility is underexplored. Why does the loss of PfVps60 reduce EVs production? Is this the only pathway responsible for EVs biogenesis?

Response: PfVps60KO is susceptible to ART as reported in Supplementary Figure 1 [RSA < 1%].

The loss of PfVps60 and the subsequent reduction in EVs production has already been established by Avalos-Padilla et al., 2021. Briefly, the authors looked for parasite homologs of proteins involved in the human endosomal sorting complex required for transport (ESCRT) of which PfVps60 was one and the only successful knock out that reduced EVs abundance.

Our observation that PfVps60KO under DHA produced comparable to wildtype parasite lines (Supplementary Figure 2C, p-value = 0.18) amounts of EVs indicates that PfVps60 is not the only pathway responsible for EVs biogenesis in Plasmodium falciparum.

The exposure of R561H and PfVps60KO strains to 700 nM DHA is not sufficiently justified.
Why were 700 nM chosen? Is this physiologically relevant?How does this compare with DHA plasma concentrations in patients receiving artemisinin-based combination therapy?

Response: The choice of 700nM DHA concentration was based on its use in the well-established in vitro ring-stage survival assay (RSA).  Line 341-342

It is not necessarily a concentration that correlates with in vivo DHA plasma concentrations in patients receiving artemisinin-based combination therapy.

"A role for EVs in the molecular mechanisms undergirding P. falciparum ART resistance has traditionally been explored within the hypothetical parasite-parasite communication motif."
"Motif" is an inappropriate term in this context. A more precise term would be "framework" or "hypothesis”.

Response: This has been revised accordingly to "A role for EVs in the molecular mechanisms undergirding P. falciparum ART resistance has traditionally been explored within the hypothetical parasite-parasite communication framework." Lines 70 - 72

Result

The resistance phenotype determination relies solely on Ring-stage Survival Assay (RSA), but there is no mention of biological replicates beyond triplicates. Given the variability in RSA, triplicates might not be sufficient to draw strong conclusions.

Response: The three biological replicates used in the experiments were 3D7, PfVps60KO and R561H parasite lines each with 3 technical replicates (Supplementary Figure 1). The number of technical replicates was informed by the minimum needed to ensure experimental validity and reliability.

The interpretation of EVs abundance under different conditions is inconsistent:
The claim "DHA treatment condition is the best predictor of EVs abundance" is based on a simple linear regression model with an R² of only 0.304 (adjusted R² = 0.177). This means only ~17% of the variation is explained by the model, which is very weak.
The p-value for DHA treatment (0.018) is modest, and there is no correction for multiple hypothesis testing, which raises the risk of false positives.

Response: The claim that "DHA treatment condition is the best predictor of EVs abundance" is made in comparison to the DMSO treatment condition (p-value = 0.96), artemisinin-susceptible parasite line (p-value = 0.6), and the artemisinin- resistant phenotype (p-value = 0.312) (Table 3). This is the context in which the adjective "best" was used.

A linear regression model with multiple variable input (parasite lines and treatment conditions) was used to determine the best predictor of EVs abundance. The model automatically adjusts the p-values for each of the predictors.

Why are DHA-treated PfVps60KO parasites producing the most EVs despite being described as deficient in EVs production? This contradicts the premise that PfVps60KO should have impaired EVs biogenesis.

Response: This seeming contradiction is made clear with the following statements.

  • PfVps60KO untreated/ DMSO has least EVs in comparison to 3D7 and R561H (Table 2 and supplementary figures 2A and 2B). This establishes the baseline that PfVps60KO has impaired EVs biogenesis.
  • Under DHA treatment PfVps60KO has the most EVs in comparison to 3D7 and R561H (Table 2 and supplementary figure 2 C). These observations suggest that there are other pathways (perhaps a yet to be identified DHA-inducible pathway) involved in EVs biogenesis. Lines 244 – 246.

The PLAAC tool for prion-like domain (PrLD) prediction is prone to false positives, and there is no independent validation of aggregation-prone behavior using biochemical assays (e.g., Thioflavin T fluorescence, Congo Red binding, or TEM imaging)

Response: This has been highlighted as a limitation to our findings. Lines 306 - 308

Was the depth of LC-MS/MS proteomics sufficient? Why were only 9 falciparum proteins detected?

Response: Yes, the depth was sufficient given the wet-lab procedures for EVs extraction and the relatively small amounts of proteins extracted from the EVs (Supplementary Table 6). Future experiments will increase the volume of culture supernatant and pool technical replicates to increase the starting protein concentration for LC-MSMS.

With respect to our detection of only 9 parasite proteins after filtering, replicating this study in the future will help contextualise this finding better.

The claim that DMSO controls should not show differences in EVs production contradicts observed differences between 3D7 and PfVps60KO in the DMSO condition.

Response: DMSO was used as a vehicle control because it was the solvent used to prepare DHA. The claim was that DMSO will not alter EVs biogenesis so that any changes seen in the DHA-treated condition could be wholly attributed to DHA. Therefore, the observations seen in Table 2 and Supplementary Figure 2 are not at variance with our expectation.

Discussion

The hypothesis predicts that ART-resistant parasites should produce more EVs under DHA to remove damaged proteins. However, the data shows that the ART-resistant strain (R561H) did not have the highest EVs levels. Instead, PfVps60KO, an ART-sensitive strain, had the most EVs under DHA exposure, which does not fit the hypothesis.

Response: That is correct. However, under DHA exposure, EVs were increased in the artemisinin-resistant line R561H (Table 2 and Supplementary Figure 2). This was discussed in lines 230 - 246. The highlight here is that our data implicates an alternative DHA-inducible EVs biogenesis pathway in P. falciparum.

Materials and Methods

No functional tests (e.g., assays showing actual protein misfolding or toxicity) were done to confirm that these proteins are harmful or need to be removed via EVs. Electron microscopy (EM) or nanoparticle tracking analysis (NTA) is not used to confirm whether the isolated particles are true EVs or cellular debris. Additionally, there is no mention of using EVs markers (e.g., CD63, TSG101, HSP70) to confirm vesicle identity.
The study relies on PLAAC software to predict aggregation-prone peptides, but no experimental validation was done.

Response: These have been listed as caveats to the interpretation of our work in the limitation section of the discussion. Lines 288 -290 and 306 – 308.

Comments on the Quality of English Language
The manuscript must be checked for English text and grammar.

Response: This has been checked and revisions made accordingly.

Reviewer 3 Report

Comments and Suggestions for Authors

In the current manuscript, the authors tried to test their EV export model of ART resistance in P. falciparum by measuring the effect of DHA treatment on EV size and abundance and the content of the EV cargoes. The authors claimed to find a strong correlation between DHA treatment and increased EV abundance, which is consistent with the EV export model. However, the authors only found one aggregation-prone protein among the EV cargo proteins, which is not consistent with the EV export model that predicts enrichment of aggregation-prone proteins in the EV.

The reviewer’s comments are listed below.

  1. If the definition of the EV export model stated in line 77-79 is correct, the authors’ may need to reconsider their expectation on the EV cargoes. The model hypothesizes that the EVs are used to help remove the damaged proteins induced by DHA-driven oxidative stress. However, the damaged proteins are not equivalent to the aggregation-prone proteins that the authors are looking for during mass spec. The reason is that the damaged proteins may not be aggregation-prone before the damage. Therefore, these proteins may not be aggregation prone by nature or by its primary sequences. Instead, the authors may want to look into proteins in the EV that have more ubiquitylation or alkylation and other modifications linked to oxidative stress.
  2. The data in Figure 2 and Supplementary figures 2-3 is weak. As the authors acknowledged, the trend between the DHA-treatment and increased EV abundance seems there but the data lacks statistical significance, which could be due to low sample size or outlier data. In this case, the reviewer suggests that authors conduct more experiments and see whether they can obtain more clear results. At the current stage, the data is not strong enough to support the authors interpretations.
  3. The protocol for EV purification can be improved. The authors used the Amicon concentrator to concentrate the vesicles. Based on the reviewer’s experience, a substantial amount of vesicles can be lost on the concentrator membrane using this method. Furthermore, so far there is no study on whether the concentrator membrane preferentially traps vesicles of certain properties, such as size and weight. Therefore the EV abundance obtained may be very different from reality. The reviewer recommends using ultracentrifugation to collect the EVs and overnight gentle rocking to resuspend the EVs for DLS measurements. The authors can refer to the protocol in doi:https://doi.org/10.1101/2024.11.25.625301.
  4. In figure 1, please label the step when DHA-treatment is introduced.
  5. The reviewer prefers that all bar plots use the same format as supplementary figure 1-3. Those look more clear than figure 2 in the current version of manuscript. In Figure 2, it is also not clear what the black dots mean. Based on the supplementary data, each group has three replicates but all black dots are plotted on the DMSO group.

Author Response

In the current manuscript, the authors tried to test their EV export model of ART resistance in P. falciparum by measuring the effect of DHA treatment on EV size and abundance and the content of the EV cargoes. The authors claimed to find a strong correlation between DHA treatment and increased EV abundance, which is consistent with the EV export model. However, the authors only found one aggregation-prone protein among the EV cargo proteins, which is not consistent with the EV export model that predicts enrichment of aggregation-prone proteins in the EV.

The reviewer’s comments are listed below.

If the definition of the EV export model stated in line 77-79 is correct, the authors’ may need to reconsider their expectation on the EV cargoes. The model hypothesizes that the EVs are used to help remove the damaged proteins induced by DHA-driven oxidative stress. However, the damaged proteins are not equivalent to the aggregation-prone proteins that the authors are looking for during mass spec. The reason is that the damaged proteins may not be aggregation-prone before the damage. Therefore, these proteins may not be aggregation prone by nature or by its primary sequences. Instead, the authors may want to look into proteins in the EV that have more ubiquitylation or alkylation and other modifications linked to oxidative stress.

Response: These have been listed as caveats to the interpretation of our work in the limitation section of the discussion. Lines 308 - 311

The data in Figure 2 and Supplementary figures 2-3 is weak. As the authors acknowledged, the trend between the DHA-treatment and increased EV abundance seems there but the data lacks statistical significance, which could be due to low sample size or outlier data. In this case, the reviewer suggests that authors conduct more experiments and see whether they can obtain more clear results. At the current stage, the data is not strong enough to support the authors interpretations.

Response: This concern has been added to the discussion in lines 312 – 323.

“We contextualise our findings that lacked statistical significance by highlighting the argument that the frequentist p-value used in null hypothesis significance testing (reject or do not reject) is disadvantaged by the arbitrary threshold of 0.05 or 0.1 used. A p-value below 0.05 is considered significant and a p-value above 0.05 is not significant even if the difference between them is minimal. This oversimplifies significance decision making as a small p-value does not conclusively mean that the effect is practically significant and only indicates that the observed data is unlikely under the null hypothesis. The converse argument holds too, as a large p-value does not conclusively mean that the effect is not significant but only indicates that the observed data is more likely under the null hypothesis. Future studies will control for these concerns by using larger sample sizes and perform statistical hypotheses testing using both the frequentist p-value and Bayesian statistics bayes factors.”

The protocol for EV purification can be improved. The authors used the Amicon concentrator to concentrate the vesicles. Based on the reviewer’s experience, a substantial amount of vesicles can be lost on the concentrator membrane using this method. Furthermore, so far there is no study on whether the concentrator membrane preferentially traps vesicles of certain properties, such as size and weight. Therefore the EV abundance obtained may be very different from reality. The reviewer recommends using ultracentrifugation to collect the EVs and overnight gentle rocking to resuspend the EVs for DLS measurements. The authors can refer to the protocol in doi:https://doi.org/10.1101/2024.11.25.625301.

Response: Acknowledged. We will make these adaptations in future iterations of this work.

In figure 1, please label the step when DHA-treatment is introduced.

Response: Revision made accordingly.

The reviewer prefers that all bar plots use the same format as supplementary figure 1-3. Those look more clear than figure 2 in the current version of manuscript. In Figure 2, it is also not clear what the black dots mean. Based on the supplementary data, each group has three replicates but all black dots are plotted on the DMSO group.

Response: Revision made accordingly.

Round 2

Reviewer 2 Report

Comments and Suggestions for Authors

The authors have addressed my queries

Comments on the Quality of English Language

NA

Author Response

Comments and Suggestions for Authors

The authors have addressed my queries

Comments on the Quality of English Language: NA

Response: Thank you.

Reviewer 3 Report

Comments and Suggestions for Authors

The revised manuscript has improved. The authors included a clear acknowledgment of the limitations of this study. The figures were also updated and are more clear. 

The review suggests that in Figure 2, the authors can plot the data points of each condition onto their corresponding box plot so that the red data points are on the red box, etc.

Author Response

Comments and Suggestions for Authors

The revised manuscript has improved. The authors included a clear acknowledgment of the limitations of this study. The figures were also updated and are more clear. 

The review suggests that in Figure 2, the authors can plot the data points of each condition onto their corresponding box plot so that the red data points are on the red box, etc.

Response: This has been revised accordingly.